# Molecular epidemiology of scrub typhus in Taiwan during 2006–2016

**Hsiang-Fei Chen[1‡], Shih-Huan Peng[1‡], Kun-Hsien Tsai[2,3], Cheng-Fen Yang[1], Mei-Chun Chang[1], Yeou-Lin Hsueh[1], Chien-Ling Su[1], Ruo-Yu Wang[1], Pei-Yun Shu[1]\*, Su-Lin Yang[1]\***

**1** Center for Diagnostics and Vaccine Development, Centers for Disease Control, Ministry of Health and Welfare, Taiwan, Republic of China, **2** Institute of Environmental and Occupational Health Sciences, College of Public Health, National Taiwan University, Taipei, Republic of China, **3** Department of Public Health, College of Public Health, National Taiwan University, Taipei, Republic of China

‡ These authors share first authorship on this work.
\* pyshu@cdc.gov.tw (P-YS); cerline@cdc.gov.tw (S-LY)

**Data Availability Statement:** All relevant data are within the manuscript and its Supporting Information files.

**Funding:** This work was supported by grants MOHW104-CDC-315-000115 (SLY), MOHW105-

## Abstract

Scrub typhus is the most common endemic vector-borne disease in Taiwan. We identified a total of 4,857 laboratory-confirmed cases during 2006–2016 with hyperendemic foci on off-shore islands, including Penghu (778 cases, 16.0%) and Kinmen (716 cases, 14.7%), and eastern Taiwan, including Taitung (628 cases, 12.9%) and Hualien (508 cases, 10.5%). Scrub typhus cases occur year-round throughout Taiwan, with a summer peak in June and July. A total of 545 *O. tsutsugamushi* isolates were successfully obtained from patients infected in diverse geographic areas, including Taiwan and three offshore islands, and the complete open reading frame of the 56 kDa type-specific antigen gene (*tsa56*) sequence of these isolates was examined. High phylogenetic diversity was found in these isolates, which could be grouped into 36 distinct sequence types. Most isolates belonged to the Karp (49.9%; 272/545), followed by the TW-22 (17.8%; 97/454) and Kawasaki (14.7%; 80/545) genotypes. In conclusion, our data indicate the widespread presence of *tsa56* genotypes closely related to Thailand and Korean strains and the presence of the unique endemic strains TW-12, TW-22, TW-29, and TW-36 in Taiwan.

## Author summary

Scrub typhus is a mite-borne disease and the most common rickettsial disease in the regions including Asia-Pacific, Middle East, and South America. Taiwan is located at the center of the endemic area of scrub typhus, and surveillance of scrub typhus and monitoring of genetic sequences of *O. tsutsugamushi* are important for disease control and prevention. In this study, we present the results of laboratory-based scrub typhus surveillance in Taiwan during 2006–2016. A total of 4,857 laboratory-confirmed scrub typhus cases were identified. Most of the cases occurred on offshore islands and in eastern Taiwan, including Penghu Island (778 cases, 16.0%), Kinmen Island (716 cases, 14.7%), Taitung County (628 cases, 12.9%) and Hualien County (508 cases, 10.5%). Phylogenetic analyses of the 56-kDa type-specific antigen gene sequence of 545 *O. tsutsugamushi* strains isolated from scrub

CDC-315-122110(SLY), and MOHW106-CDC-C-315-113112(SLY) from Centers for Disease Control, Ministry of Health and Welfare, Taiwan, Republic of China. The funder had no role in study design, data collection and analysis, decision to publish, or preparation of the manuscript.

**Competing interests:** The authors have declared that no competing interests exist.

typhus patients were conducted. Most isolates belonged to the Karp (49.9%; 272/545) genotype, followed by the TW-22 (17.8%; 97/454) sequence type and Kawasaki (14.7%; 80/545) genotype. Most strains were closely related to strains from Southeast and East Asia, whereas unique isolates were also found in Taiwan.

## Introduction

Scrub typhus is an acute febrile illness caused by the obligate intracellular bacterium *Orientia tsutsugamushi* contracted from the bite of an infected larval-stage trombiculid mite (chigger) [1]. Scrub typhus is endemic to the Asia-Pacific region in an area known as the "tsutsugamushi triangle", Taiwan being located in the center. Globally, it has been estimated that one billion people are at risk of scrub typhus and that one million infections occur every year [2,3]. Symptoms of scrub typhus include fever, headache, rash, eschar, cough, myalgias, nausea, vomiting, and abdominal pain. Severe manifestations may include pneumonitis, meningitis, encephalitis, disseminated intravascular coagulation, septic shock, myocarditis, and multiorgan failure [4–6]. The case fatality rate can be up to 30–70% if not treated appropriately [7–9]. Currently, there is no commercially available vaccine.

Scrub typhus is the most common rickettsial disease in Taiwan and has been designated a notifiable infectious disease since 1955. Blood and serum samples from suspected scrub typhus patients are sent to the Taiwan Centers for Disease Control (Taiwan CDC) for diagnosis. Serological diagnosis has been performed using indirect immunofluorescent assay (IFA) since 1955, using paired sera when available. Since 2006, molecular diagnosis has been performed using real-time polymerase chain reaction (qPCR) for rapid testing and increased sensitivity with whole-blood specimens, from which bacterial isolation is also routinely performed. Scrub typhus cases are reported weekly through the National Notifiable Disease Surveillance System (nidss.cdc.gov.tw), and the number of laboratory-confirmed cases of scrub typhus exceeds 300 cases annually.

The immunodominant 56 kDa type-specific antigen gene (*tsa56*) of *O. tsutsugamushi* has been the most widely used gene target for phylogenetic analysis because of its sequence variation [10–13]. Previous studies have revealed high phylogenetic diversity among *tsa56* genotypes in Taiwan [11, 14]. We described 116 clinical isolates and 22 distinct *tsa56* sequence types during 2006–2007 [11]. Currently, the genotypic diversity and molecular epidemiology of *O. tsutsugamushi* remain unclear, including the distribution of *tsa56* genotypes by region, its seasonality, and the clinical manifestations associated with infection by different *tsa56* genotypes. Here, we studied the *tsa56* genotypes of 545 *O. tsutsugamushi* clinical isolates throughout Taiwan, including offshore islands, from 2006 to 2016 to elucidate the molecular epidemiology of scrub typhus in Taiwan.

## Materials and methods

### Ethics statement

The study protocol was approved by the Taiwan Centers for Disease Control Institutional Review Board (IRB 106111).

### Human blood samples

Isolates in this study were obtained from blood samples of confirmed cases of scrub typhus infection from 2006 to 2016. Sample data were depersonalized for anonymity. Samples were

considered positive for scrub typhus with a positive real-time polymerase chain reaction (PCR) test or IFA test, indicated by a $\geq$ 4-fold increase in *O. tsutsugamushi*-specific immunoglobulin M (IgM) or IgG antibody in paired sera.

## DNA extraction and real-time PCR

Peripheral blood mononuclear cells (PBMCs) were purified from 4 mL of whole blood samples using Ficoll-Paque Plus (GE Health care Bio-Sciences AB, Uppsala, Sweden) according to the manufacturer's instructions, washed and resuspended in 400 μL of phosphate-buffered saline (PBS) containing 2% fetal calf serum. DNA extraction was performed with the QIAamp DNA Blood Mini Kit (QIAGEN GmbH, Hilden, Germany) according to the manufacturer's instructions with 200 μL of each PBMC suspension. Two SYBR-based qPCR assays were used to test for *O. tsutsugamushi*, targeting *tsa56* (RST-14F: 59-CCATTTGGTGG TACATTAGCTGCA GGT-39; RST-6R: 59-TCACGATCAGC TATACTTATAGGCA-39) and the 16S ribosomal RNA gene (*rrs*) (OTF7: 59-CCAGYGGGTRATGCCGGGAACTAT-39; OTR6: 59GGCAGT GTGTACAAGGCCCGAGAA-39), performed using the Fast Start Essential DNA Green Master kit (Roche Diagnostics, Basel, Switzerland). Samples were considered positive if both targets were amplified.

## Isolation of O. tsutsugamushi

PBMCs collected from acute-phase blood samples of scrub typhus patients were used for isolation of *O. tsutsugamushi*. Bacterial isolation in cell culture was performed using the centrifugation shell vial technique as described previously [15,16]. Briefly, *O. tsutsugamushi* was propagated in L929 mouse fibroblast cells (ATCC CCL-1, NCTC Clone 929) at 32˚C in a 5% CO2 incubator for 10 to 14 days and then detected by IFA using an *O. tsutsugamushi*-specific antibody. Each positive shell vial was harvested and inoculated into a T-25 flask containing a monolayer of confluent L929 cells. After 14–20 days of incubation, the bacteria-infected L929 cells were scraped up and frozen at −80˚C. Isolated bacteria were identified using the nomenclature OT/country of origin/strain/year of isolation.

## Indirect IFA

*O. tsutsugamushi* whole-cell antigens Karp, Gilliam, and Kato strains dotted on Teflon-coated spot glass slides were used for IFA as previously described [17]. Briefly, whole-cell antigens were fixed and permeabilized with ice-cold acetone/methanol (1:1) for 10 minutes, and the slides were air-dried and blocked with PBS containing 1% goat serum. Serum samples were serially diluted and incubated in a humidified atmosphere for 30 minutes at 37˚C. Subsequently, fluorescein isothiocyanate–conjugated anti-human IgM and IgG (Sigma, St. Louis, MO, USA) were diluted with PBS containing Evans blue counterstain (Sigma Chemical Company, St. Louis, MO, USA) and applied to an antigen-coated spot in a humidified atmosphere for 30 minutes at 37˚C. The slides were examined by epifluorescence microscopy (Zeiss, Axio Imager 2, Jena, Germany) by two observers at a magnification of ×400. The binding endpoint titer was determined as the highest dilution with a positive fluorescence reaction.

## PCR amplification and nucleotide sequencing

PCR and DNA sequencing of the complete *tsa56* ORF was performed as previously described [17]. Briefly, bacterial DNA was extracted from *O. tsutsugamushi*-infected L929 cells using the QIAamp DNA Blood Mini Kit (QIAGEN GmbH, Hilden, Germany) according to the manufacturer's instructions and stored at -80˚C. Primers used for PCR and nucleotide sequencing

were as previously described [17]. PCR amplification was performed in 50 μL volumes using the QIAGEN Taq PCR Core Kit (QIAGEN, Hilden, Germany) according to the manufacturer's protocol. PCR products were purified using a QIAQuick Gel Extraction Kit (QIAGEN, Hilden, Germany). Nucleotide sequences were determined by an automated DNA sequencing kit and an ABI Prism 3730XL DNA sequencer (Applied Biosystems, Foster City, CA, USA) according to the manufacturer's protocols. Overlapping nucleotide sequences were combined for analysis and edited with the Lasergene software package (DNASTAR Inc, Madison, WI, USA). Nucleotide sequences were submitted to GenBank. The strain identifiers and their accession numbers are listed in S1 Table.

### Phylogenetic analysis

Complete *tsa56* nucleotide sequences obtained in this study were aligned with global *tsa56* sequences retrieved from GenBank using Clustal W software. Phylogenetic analysis was conducted using MEGA version 7 (http://www.megasoftware.net/) [18] with the neighbor-joining method and the maximum composite likelihood as a substitution model. One thousand bootstrap replicates were performed to estimate the node reliability of the phylogenetic tree, and bootstrap support values above 75 were considered significant.

## Results

### Epidemiology of scrub typhus in Taiwan 2006–2016

A total of 28,626 suspected cases were sent to the Taiwan CDC for confirmation of scrub typhus infection 2006–2016; among them, 4,857 cases were confirmed by laboratory diagnosis. Fig 1 shows the annual numbers of confirmed cases of scrub typhus and the incidence per 100,000 persons during 2006–2016. Confirmed cases of scrub typhus were between 322 and 538 per year. Fig 2 shows the geographic distribution of the scrub typhus cases in Taiwan and the offshore islands. Penghu County had the highest number of scrub typhus cases (778 cases; 16.0%), followed by Kinmen County (716 cases; 14.7%) and Taitung County (628 cases;

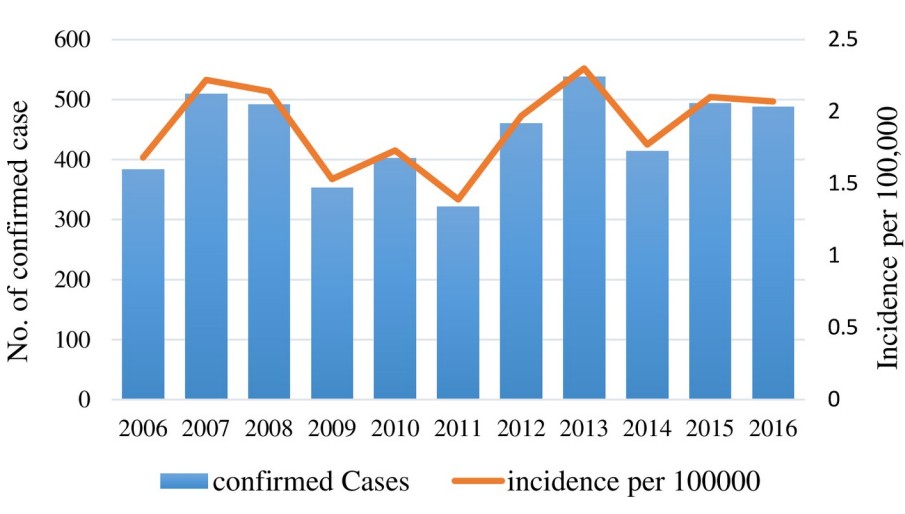

| Year | Confirmed Cases | Incidence per 100000 |
|------|-----------------|----------------------|
| 2006 | 384 | 1.68 |
| 2007 | 510 | 2.22 |
| 2008 | 492 | 2.14 |
| 2009 | 353 | 1.53 |
| 2010 | 402 | 1.74 |
| 2011 | 322 | 1.39 |
| 2012 | 460 | 1.97 |
| 2013 | 538 | 2.30 |
| 2014 | 414 | 1.77 |
| 2015 | 494 | 2.10 |
| 2016 | 488 | 2.07 |
| total | 4857 | mean : 1.90 |

**Fig 1. Annual numbers of confirmed cases of scrub typhus cases in Taiwan 2006–2016.** There is an increasing trend of *O. tsutsugamushi* infection in Taiwan.

**A.**                                    **B.**

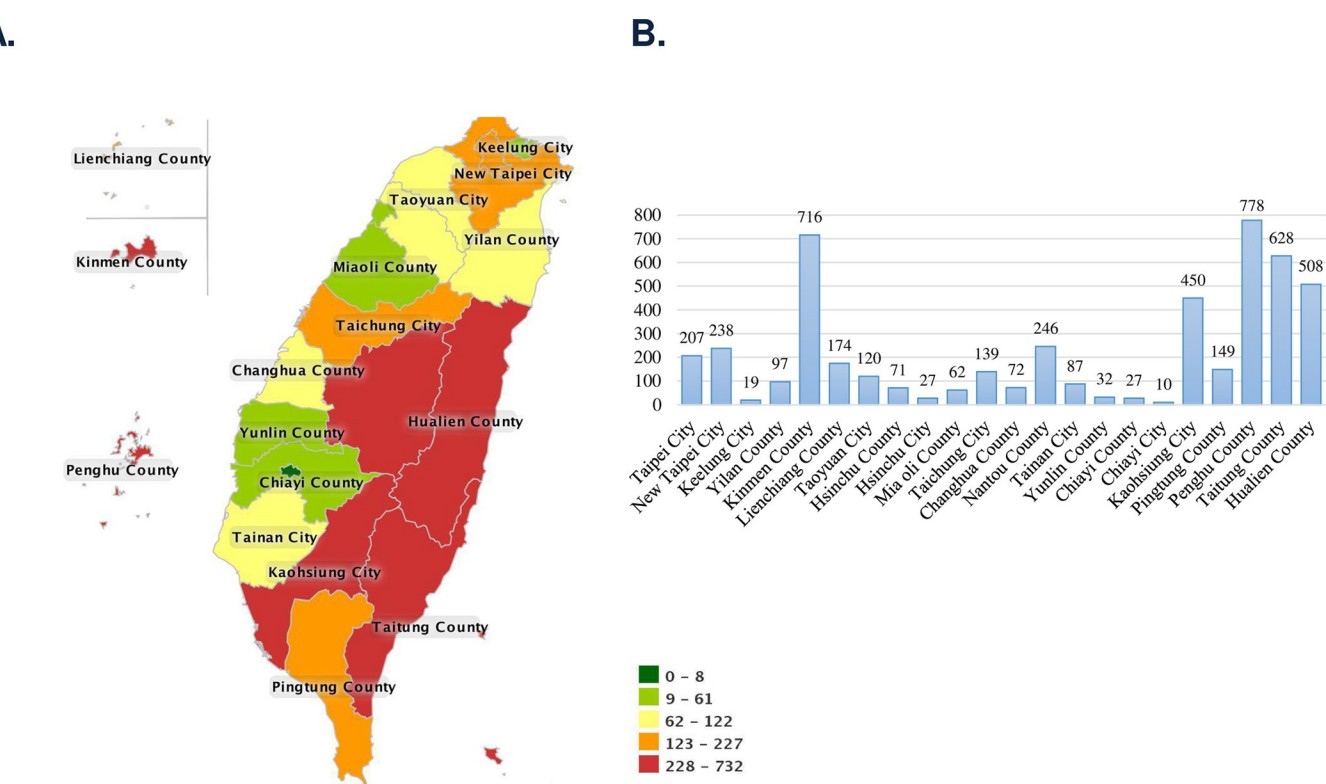

**Fig 2. Geographic distribution of scrub typhus cases in Taiwan 2006–2016.** Most cases were identified in eastern Taiwan (Hualien and Taitung Counties) and on offshore islands (Penghu and Kinmen Counties). The geographic map was acquired from the Taiwan Centers for Disease Control Open Data Platform (https://nidss.cdc.gov.tw).

12.9%). Fig 3 shows the monthly distribution of scrub typhus cases. Scrub typhus occurs throughout the year in Taiwan. There were two peaks, a major peak in July and a small peak in October. Fig 4 shows the gender and age distribution of the confirmed cases. A total of 3,027 cases (62.4%) were male, and 1,825 (37.6%) cases were female, with a male-to-female ratio of 1.66:1. There were 1042 cases (21%) that occurred in 50–59 years old, 848 cases (19.5%) in 40–49 years old, and 45.3% of cases occurred in those older than 50 years old, and 3.6% of cases occurred among those under 9 years old.

### Phylogenetic analysis of O. tsutsugamushi strains

A total of 545 *O. tsutsugamushi* isolates were obtained from scrub typhus cases during 2006–2016 with *tsa56* sequences. The phylogenetic analysis classified these isolates into 36 distinct sequence types according to their sequence similarities being higher than 98% (Fig 5). The accession numbers of the identified strains are listed in S1 Table. Phylogenetic analysis revealed that most isolates were grouped into the Karp genotype, including TW-1 to TW-8, TW-23 to TW-26, and TW-31 to TW-33, which are closely related to *tsa56* sequences from Thailand, Korea, Cambodia, and New Guinea (Table 1). Strains TW-13 to TW-19 and TW-30 were similar to the Kawasaki genotype and were closely related to *tsa56* sequences from Thailand, Japan and China, respectively. TW-9 was identified as the Kuroki genotype and was most closely related to Boryong and Kuroki strains from Korea and Japan, respectively. TW-10 to TW-11 and TW-27 to TW-28 were clustered with TA763-type strains with sequence similarity to the TA763 strain isolated from Thailand and Vietnam. TW-20 and TW-21 belong to

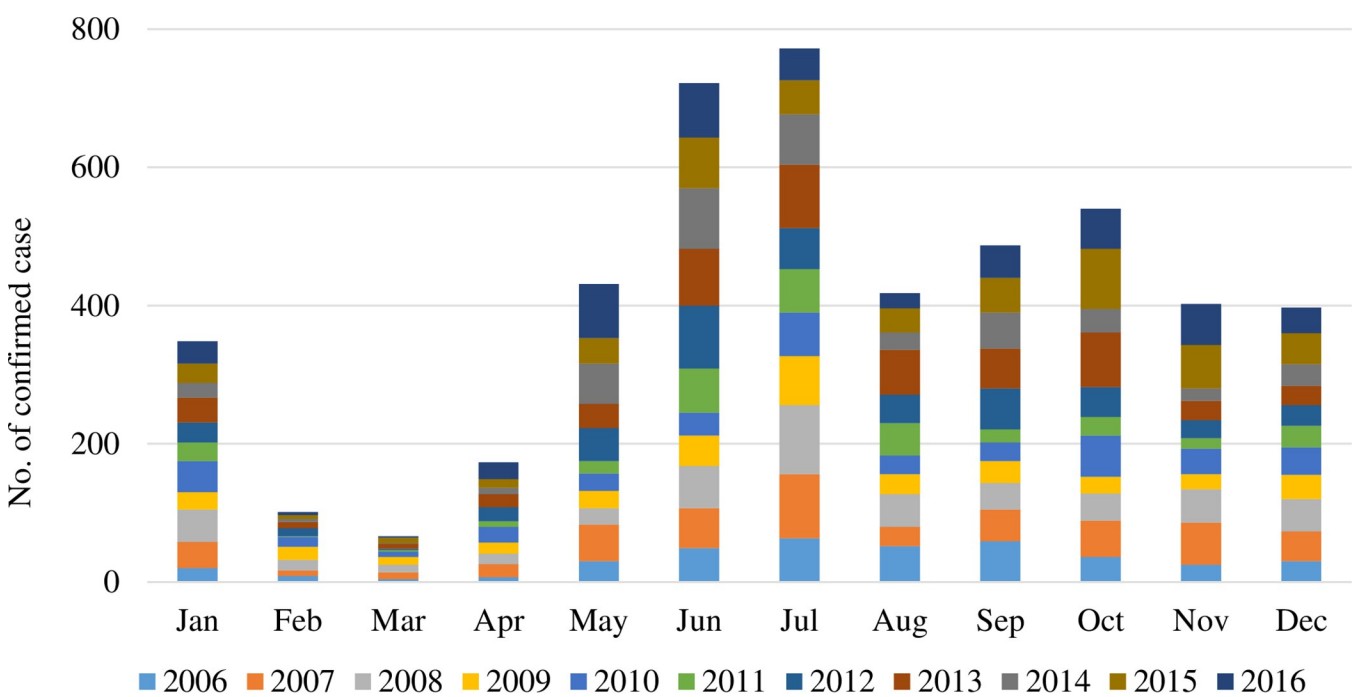

| Monthly distribution from 2006-2016 | | | | | | | | | | | |
|---|---|---|---|---|---|---|---|---|---|---|---|
| Month | 1 | 2 | 3 | 4 | 5 | 6 | 7 | 8 | 9 | 10 | 11 | 12 |
| Cases | 348 | 101 | 66 | 173 | 431 | 722 | 772 | 418 | 487 | 540 | 402 | 397 |

**Fig 3. Monthly distribution of scrub typhus cases in Taiwan 2006–2016.** Scrub typhus occurs year-round and peaks in the spring and fall seasons.

the Kato genotype and were most closely related to strains from Malaysia and Japan, respectively. TW-34 was closely related to the Gilliam strain isolated in Cambodia. We found that TW-12, TW-22, TW-29, and TW-36 were unique in Taiwan and also distinct from strains isolated from other countries.

## Geographic distribution of O. tsutsugamushi isolates

Table 2 shows the geographic distribution of *O. tsutsugamushi* isolates in Taiwan. TW-1 represents the most abundant *O. tsutsugamushi* isolates in Taiwan, especially on offshore islands. Forty-eight isolates were grouped into TW-9, distributed in the northern, central and eastern parts of Taiwan but not in southern Taiwan or the offshore islands. TW-19 and TW-22 contained 54 and 97 isolates, respectively, and were widely distributed on Taiwan's main island and the Kinmen and Lienchiang islands.

In this study, TW-11, TW-13, and TW-33 to TW-35 were found in Central Taiwan, whereas TW-15 and TW-18 were restricted to Southern Taiwan. TW-12, TW-14, and TW-30 were common in Eastern Taiwan, and TW-5 and TW-23 were discovered only on Kinmen Island. Taken together, most of the isolates came from Kinmen Island (*n* = 164). Kaohsiung City had the second-highest number of isolates (*n* = 72), TW-22 was the major sequence type, and a variety of sequence types were also found in this city.

**A.**

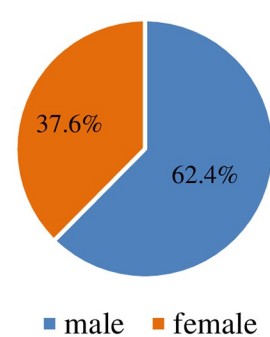

**B.**

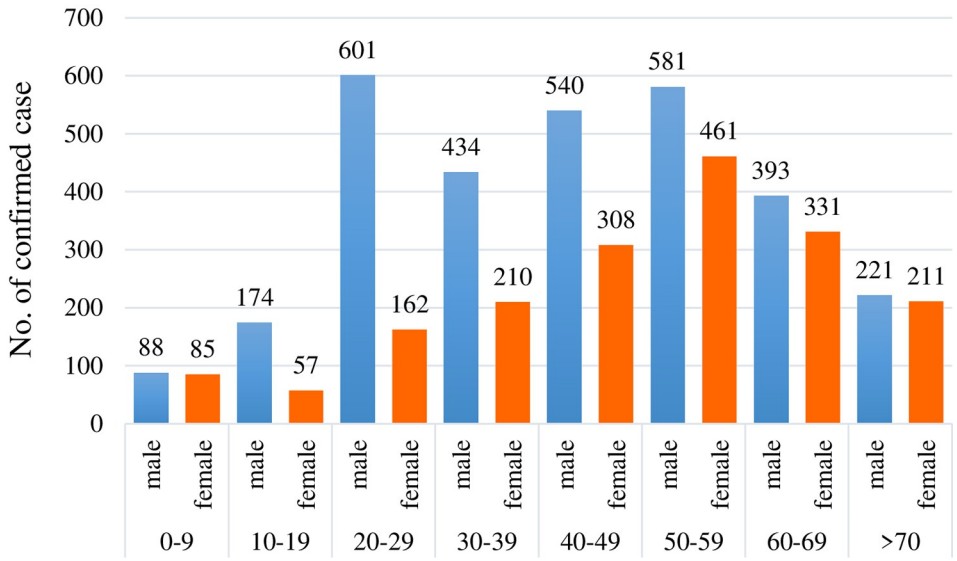

**Fig 4. Gender and age distribution of confirmed cases in Taiwan 2006 to 2016.**

### Monthly distribution of O. tsutsugamushi sequence types

The monthly distribution of *O. tsutsugamushi* sequence types is shown in Table 3. Most of the cases of scrub typhus occurred in the warm season (April to October), representing most sequence types. On the other hand, a few sequence types, including TW-9, TW-14, and TW-15, occurred in the cold season (November to February). It is worth noting that TW-22 was widely found from March to December. Nevertheless, TW-12, TW-29, and TW-36 are only found from May to September. Taken together, scrub typhus occurs throughout the year, and *O. tsutsugamushi* shows great genotype diversity.

### Clinical symptoms of the sequence types

The clinical symptoms of the patients are shown in Table 4. The most frequent symptoms included fever (89.5%, 488/545), headache (28.6%, 156/545), rashes (25.3%, 138/545), eschar (25.0%, 136/545), lymphadenopathy (9.5%, 52/545) and liver dysfunction (7.9%, 43/545). Severe manifestations, including sepsis, pneumonia, liver, and kidney dysfunction, were also observed in some patients. Sepsis was observed in TW-1, TW-22, and TW-24. Kidney

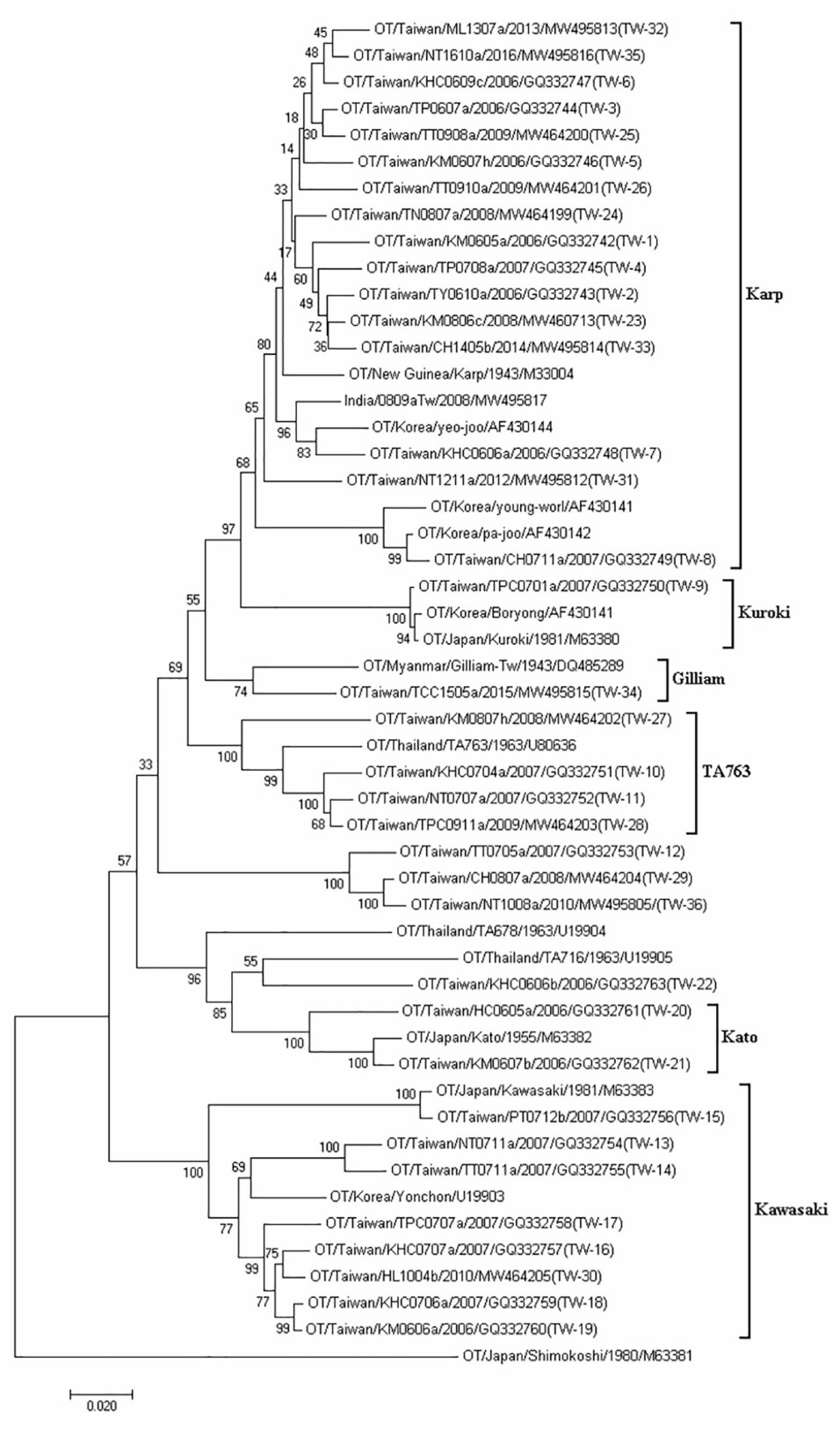

**Fig 5. Phylogenetic tree based on the 56 kDa TSA gene ORF of TW-1 to TW-36.**

**Table 1. The sequence type, representative strains, and phylogenetically closest foreign strains of *Orientia tsutsugamushi*.**

| Sequence type | Representative isolate | Length of ORF of gene | Isolation site of representative isolate | Isolation date (month/year) of representative isolate | Pairwise nucleotide sequence similarity (%) to phylogenetically closest foreign O. tsutsugamushi strain from NCBI | GenBank accession no. | Genotype | No. strain (2006–2016) |
|---|---|---|---|---|---|---|---|---|
| TW-1 | KM0605a | 1608 | Kinmen Island | 05/2006 | 98.3% to UT150 (EF213086), Thailand | GQ332742 | Karp | 198 |
| TW-2 | TY0610a | 1605 | Taoyuan County | 10/2006 | 97.4% to UT336 (EF213089), Thailand | GQ332743 | Karp | 4 |
| TW-3 | TP0607a | 1605 | Taipei County | 07/2006 | 97.5% to Karp (M33004), New Guinea | GQ332744 | Karp | 1 |
| TW-4 | TP0708a | 1608 | Taipei County | 08/2007 | 96.0% to UT336 (EF213089), Thailand | GQ332745 | Karp | 10 |
| TW-5 | KM0607h | 1632 | Kinmen Island | 07/2006 | 95.8% to UT176 (EF213081), Thailand | GQ332746 | Karp | 9 |
| TW-6 | KHC0609c | 1608 | Kaohsiung City | 09/2006 | 97.7% to UT176 (EF213081), Thailand | GQ332747 | Karp | 6 |
| TW-7 | KHC0606a | 1608 | Kaohsiung City | 06/2006 | 96.4% to yeo-joo (AF430144), Korea | GQ332748 | Karp | 6 |
| TW-8 | CH0711a | 1692 | Changhua County | 11/2007 | 96.3% to pa-joo (AF430142) Korea | GQ332749 | Karp | 7 |
| TW-9 | TPC0701a | 1599 | Taipei City | 01/2007 | 99.7% to Boryong (AM494475), Korea | GQ332750 | Kuroki | 48 |
| TW-10 | KHC0704a | 1566 | Kaohsiung City | 04/2007 | 93.8% to TA763 (U80636), Thailand | GQ332751 | TA763 | 18 |
| TW-11 | NT0707a | 1584 | Nantou County | 07/2007 | 96.7% to TA763 (U80636), Thailand | GQ332752 | TA763 | 1 |
| TW-12 | TT0705a | 1593 | Taitung County | 05/2007 | 86.8% to UT302 (EF213095), Thailand | GQ332753 | TW-12 | 2 |
| TW-13 | NT0711a | 1557 | Nantou County | 11/2007 | 92.6% to Sxh951 (AF050669), China | GQ332754 | Kawasaki | 3 |
| TW-14 | TT0711a | 1551 | Taitung County | 11/2007 | 92.6% to Ikeda (AP008981), Japan | GQ332755 | Kawasaki | 1 |
| TW-15 | PT0712b | 1569 | Pingtung County | 12/2007 | 99.3% to Kawasaki (M63383), Japan | GQ332756 | Kawasaki | 1 |
| TW-16 | KHC0707a | 1572 | Kaohsiung City | 07/2007 | 97.2% to UT329 (EF213099), Thailand | GQ332757 | Kawasaki | 11 |
| TW-17 | TPC0707a | 1596 | Taipei City | 07/2007 | 97.7% to UT125 (EF213096), Thailand | GQ332758 | Kawasaki | 2 |
| TW-18 | KHC0706a | 1596 | Kaohsiung City | 06/2007 | 98.4% to UT125 (EF213096), Thailand | GQ332759 | Kawasaki | 5 |
| TW-19 | KM0606a | 1572 | Kinmen Island | 06/2006 | 97.2% to UT125(EF213096), Thailand | GQ332760 | Kawasaki | 54 |
| TW-20 | HC0605a | 1572 | Hsinchu County | 05/2006 | 99.9% to LF-1(AF173050), Malaysia | GQ332761 | Kato | 5 |
| TW-21 | KM0607b | 1590 | Kinmen island | 07/2006 | 98.6% to Kato (M63382), Japan | GQ332762 | Kato | 6 |
| TW-22 | KHC0606b | 1575 | Kaohsiung City | 06/2006 | 88.3% to FPW1038 (EF213087), Thailand | GQ332763 | TW-22 | 97 |
| TW-23 | KM0806c | 1611 | Kinmen island | 06/2008 | 97.7% to S0902151-KH (HQ718422), Cambodia | MW460713 | Karp | 4 |
| TW-24 | TN0807a | 1602 | Tainan City | 07/2008 | 97.8% to UT336 (EF213089), Thailand | MW464199 | Karp | 12 |
| TW-25 | TT0908a | 1605 | Taitung County | 08/2009 | 96.6% to UT336 (EF213089), Thailand | MW464200 | Karp | 3 |
| TW-26 | TT0910a | 1605 | Taitung County | 10/2009 | 97.6% to UT176 (EF213081), Thailand | MW464201 | Karp | 6 |
| TW-27 | KM0807h | 1605 | Kinmen island | 07/2008 | 97.6% to 45QN-VN (HQ817459), Vietnam | MW464202 | TA763 | 5 |

(*Continued*)

**Table 1.** (Continued)

| Sequence type | Representative isolate | Length of ORF of gene | Isolation site of representative isolate | Isolation date (month/year) of representative isolate | Pairwise nucleotide sequence similarity (%) to phylogenetically closest foreign *O. tsutsugamushi* strain from NCBI | GenBank accession no. | Genotype | No. strain (2006–2016) |
|---|---|---|---|---|---|---|---|---|
| TW-28 | TPC0911a | 1587 | Taipei City | 11/2009 | 96.5% to 02QNg-VN (HQ817449) Vietnam | MW464203 | TA763 | 3 |
| TW-29 | CH0807a | 1575 | Changhua County | 07/2008 | 87.0% to UT302 (EF213095), Thailand | MW464204 | TW-29 | 4 |
| TW-30 | HL1004b | 1572 | Hualien County | 04/2010 | 98.4% to UT329(EF213099), Thailand | MW464205 | Kawasaki | 3 |
| TW-31 | NT1211a | 1602 | Nantou County | 11/2012 | 94.9% toUT219 (EF213100), Thailand | MW495810 | Karp | 2 |
| TW-32 | ML1307a | 1599 | Miaoli County | 07/2013 | 96.4% to UT177 (EF213084), Thailand | MW495812 | Karp | 2 |
| TW-33 | CH1405b | 1608 | Changhua County | 05/2014 | 99.2% to S0902151_KH HQ718422, Cambodia | MW495814 | Karp | 1 |
| TW-34 | TCC1505a | 1599 | Taichung City | 05/2015 | 95.1% to S0617100_KH (HQ718421) Cambodia | MW495815 | Gilliam | 1 |
| TW-35 | NT1610a | 1608 | Nantou County | 10/2016 | 97.5% to UT176 (EF213081), Thailand | MW495816 | Karp | 1 |
| TW-36 | NT1008a | 1581 | Nantou County | 08/2010 | 86.5% to UT302 (EF213095), Thailand | MW495805 | TW-36 | 3 |
| India/0809aTw | | 1599 | India | 09/2008 | 98.9% to UT219 (EF213100), Thailand | MW495817 | Karp | 1 |

dysfunction was observed in TW-1 and TW-30. Consciousness changes were caused by TW-1 and TW-22. Overall, patients infected with TW-1 and TW-22 displayed more complicated syndromes and severe illness in Taiwan during the 2006–2016 surveillance.

## Discussion

Scrub typhus was first reported in 1908 [19]. There are approximately 350 scrub typhus cases in Taiwan annually. In the present epidemiological study, we analyzed 4875 human cases collected from 2006–2016 and found that eastern Taiwan and the offshore islands displayed a higher prevalence. In particular, offshore islands accounted for 34.3% of the total cases, with most outbreaks occurring in rural regions. These results demonstrated that disease transmission was highly associated with seasonal characteristics [14]. The prevalent period in early spring to late fall follows the adult mites' prosperous growth and breeding season. In addition, people often take vacation trips during summer seasons, thereby being exposed to the infected chiggers and acquiring scrub typhus. Men have a higher rate of typhus than women, possibly reflecting that men are more frequently exposed to the chiggers' living environments.

In this study, we isolated 545 indigenous *O tsutsugamushi* strains from acute-phase blood samples of scrub typhus cases and analyzed the gene sequences of the TSA56 protein. The TSA56 gene sequences were classified into 36 sequence types. Most of these isolates were closely related to strains from the southern region of the endemic area (Thailand, Vietnam, Malaysia, and New Guinea), while others were closely related to the northern region of the endemic area (northern China, Japan, and Korea). Notably, four sequence types, TW-12, TW-22, TW-29, and TW-36, are unique in Taiwan; it may be worthwhile to obtain their entire genome sequence and determine their epidemiology and phylogeny in the future.

Taiwan is an island off the southeastern coast of mainland China in the western Pacific Ocean. Taiwan is a mountainous island with one-third of the area over 1000 meters high and

**Table 2. Geographic distribution of sequence types.**

| sequence | North | | | | | | Central | | | | | South | | | | East | | Offshore | | | Total |
|---|---|---|---|---|---|---|---|---|---|---|---|---|---|---|---|---|---|---|---|---|---|
| type | YL | KL | TP | NTC | TY | HC | ML | TC | CH | NT | WL | CY | TN | KH | PT | TT | HL | KM | PH | LC | |
| TW-1 | | | 1 | 8 | 7 | | 1 | 3 | 1 | | | | 1 | 8 | 4 | 23 | 8 | 100 | 13 | 20 | 198 |
| TW-2 | | | | 1 | 1 | | | | | | | | | | | 2 | | | | | 4 |
| TW-3 | | | 1 | | | | | | | | | | | | | | | | | | 1 |
| TW-4 | | 1 | | 1 | | | | | | 1 | | | | | | | 4 | 3 | | | 10 |
| TW-5 | | | | | | | | | | | | | | | | | | 9 | | | 9 |
| TW-6 | | | | | | | | | | 1 | | | | 4 | 1 | | | | | | 6 |
| TW-7 | | | | | | | | | 1 | | | | | 3 | 1 | 1 | | | | | 6 |
| TW-8 | | | | | | | | | 1 | | | | | 1 | 1 | | | | 4 | | 7 |
| TW-9 | | 2 | 4 | 6 | 5 | 4 | 6 | 6 | | 8 | | | | | | 4 | 3 | | | | 48 |
| TW-10 | | 1 | 1 | 2 | 1 | | | | | | | | | 3 | | 6 | 4 | | | | 18 |
| TW-11 | | | | | | | | | 1 | | | | | | | | | | | | 1 |
| TW-12 | | | | | | | | | | | | | | | | 2 | | | | | 2 |
| TW-13 | | | | | | | | | | 3 | | | | | | | | | | | 3 |
| TW-14 | | | | | | | | | | | | | | | | 1 | | | | | 1 |
| TW-15 | | | | | | | | | | | | | | | 1 | | | | | | 1 |
| TW-16 | | | | 1 | | | | 1 | | | | | | 1 | | 3 | | | 1 | 4 | 11 |
| TW-17 | | | 1 | | | | | | | | | | | | | | | 1 | | | 2 |
| TW-18 | | | | | | | | | | | | | | 4 | 1 | | | | | | 5 |
| TW-19 | | | 1 | 1 | | | 1 | 3 | 3 | 10 | | | 1 | 4 | 1 | 5 | 13 | 9 | | 2 | 54 |
| TW-20 | | 1 | | | 1 | | | | | | | | | 1 | | | 2 | | | | 5 |
| TW-21 | | | | 1 | | | | | | | | | | | | | | 5 | | | 6 |
| TW-22 | | | | 3 | 2 | 1 | | 1 | 1 | | | 1 | 3 | 40 | 3 | 8 | 3 | 30 | | 1 | 97 |
| TW-23 | | | | | | | | | | | | | | | | | | 4 | | | 4 |
| TW-24 | | | | 1 | 2 | | | 1 | | | | | 2 | 2 | | 4 | | | | | 12 |
| TW-25 | | | | | | | | | | 2 | | | | | | 1 | | | | | 3 |
| TW-26 | | | 1 | | | | | | | | | | | | | 1 | 4 | | | | 6 |
| TW-27 | | | | | | | | | | | | | | 1 | | 1 | | 2 | | 1 | 5 |
| TW-28 | | | 1 | | | | | | | | | | | | | 2 | | | | | 3 |
| TW-29 | | | | 1 | 1 | | | 1 | | | | | | | | 1 | | | | | 4 |
| TW-30 | | | | | | | | | | | | | | | | | 3 | | | | 3 |
| TW-31 | | | | | | | | | | | | | 1 | | | | 1 | | | | 2 |
| TW-32 | | | | | | | 1 | | | | | 1 | | | | | | | | | 2 |
| TW-33 | | | | | | | | | 1 | | | | | | | | | | | | 1 |
| TW-34 | | | | | | | | 1 | | | | | | | | | | | | | 1 |
| TW-35 | | | | | | | | | | 1 | | | | | | | | | | | 1 |
| TW-36 | | | | | | | | | | 1 | | 1 | | | | | | 1 | | | 3 |
| Sum | 0 | 5 | 11 | 24 | 20 | 7 | 9 | 16 | 8 | 28 | 1 | 3 | 8 | 72 | 13 | 65 | 45 | 164 | 18 | 28 | 545 |

with more than two hundred peaks over 3000 meters. The range of landscapes and topography is varied and complicated in Taiwan. The climate varies with altitude, and the ecological environment is a complex system that is rich in flora and fauna [20]. The complex environment has been suggested to support the evolution and diversity of *O. tsutsugamushi* in Taiwan. Our study revealed that specific districts have dominant genotypes of *O. tsutsugamushi*. This may be ascribed to the variation of the vectors, at least in part, being controlled by geographical and seasonal factors. Wang and colleagues reported various chiggers linked to geographical and

**Table 3. Monthly distribution of *Orientia tsutsugamushi* sequence types in Taiwan.**

| Sequence type | Jan | Feb | Mar | Apr | May | Jun | Jul | Aug | Sep | Oct | Nov | Dec | Total |
|---|---|---|---|---|---|---|---|---|---|---|---|---|---|
| TW-1 | | | | 7 | 20 | 47 | 59 | 16 | 18 | 15 | 15 | 1 | 198 |
| TW-2 | | | | | 1 | 1 | | | 1 | 1 | | | 4 |
| TW-3 | | | | | | | 1 | | | | | | 1 |
| TW-4 | | | | 1 | 2 | 1 | 2 | 2 | | 1 | 1 | | 10 |
| TW-5 | | | | | | 1 | 5 | 1 | 1 | 1 | | | 9 |
| TW-6 | | | | 1 | | | 1 | 1 | 1 | 1 | 1 | | 6 |
| TW-7 | | | | | | 1 | 3 | 1 | | | | 1 | 6 |
| TW-8 | | | | | | | 2 | | | 1 | 4 | | 7 |
| TW-9 | 12 | 6 | 1 | | | | | | | | 7 | 22 | 48 |
| TW-10 | | | | | 2 | 3 | 3 | 2 | 3 | 1 | 1 | 3 | 18 |
| TW-11 | | | | | | | | 1 | | | | | 1 |
| TW-12 | | | | | 1 | 1 | | | | | | | 2 |
| TW-13 | | | | | | 1 | 1 | | | | 1 | | 3 |
| TW-14 | | | | | | | | | | | 1 | | 1 |
| TW-15 | | | | | | | | | | | | 1 | 1 |
| TW-16 | | 1 | | | | 1 | 3 | | 4 | 1 | 1 | | 11 |
| TW-17 | | | | | | | 2 | | | | | | 2 |
| TW-18 | | | | | | 1 | 1 | | 1 | | 2 | | 5 |
| TW-19 | | | 3 | 5 | 4 | 6 | 11 | 4 | 9 | 1 | 10 | 1 | 54 |
| TW-20 | | | | | 2 | | 1 | 1 | 1 | | | | 5 |
| TW-21 | | | | | | 1 | 2 | | 2 | 1 | | | 6 |
| TW-22 | | | 1 | 2 | 8 | 15 | 20 | 12 | 18 | 14 | 5 | 2 | 97 |
| TW-23 | | | | | | 2 | 1 | | 1 | | | | 4 |
| TW-24 | | | | 1 | 5 | 1 | 3 | | 1 | 1 | | | 12 |
| TW-25 | | | | | | | | 1 | | 1 | 1 | | 3 |
| TW-26 | | | | | | | 1 | | 1 | 2 | 2 | | 6 |
| TW-27 | | | | | | | 1 | | 2 | 2 | | | 5 |
| TW-28 | | | | | | | | 1 | | | 2 | | 3 |
| TW-29 | | | | | | 1 | 2 | | | 1 | | | 4 |
| TW-30 | | | | | 1 | 1 | | | | | 1 | | 3 |
| TW-31 | | | | | | | | | 1 | | 1 | | 2 |
| TW-32 | | | | | | | 1 | 1 | | | | | 2 |
| TW-33 | | | | | 1 | | | | | | | | 1 |
| TW-34 | | | | | 1 | | | | | | | | 1 |
| TW-35 | | | | | | | | | | 1 | | | 1 |
| TW-36 | | | | | | | | 2 | 1 | | | | 3 |
| Sum | 12 | 7 | 5 | 19 | 50 | 87 | 122 | 51 | 60 | 46 | 55 | 31 | 545 |

seasonal variation that act as potential vectors, leading to the formation of a dominant *O. tsutsugamushi* transmission chain in Taiwan [21]. In addition, invasive plants change the survival of certain vectors and affect the transmission of *O. tsutsugamushi* [22].

Continuous surveillance and analysis of TSA56 gene sequences may have beneficial effects in epidemiology and public health research on scrub typhus. In this study, scrub typhus was found with two sharp peaks in Taiwan, from May to July and September to November. Most *O. tsutsugamushi* isolates are obtained in the warm season (April to October), and their *tsa56* sequences are closely related to strains identified in Southeast Asia. Nevertheless, TW-9, TW-

**Table 4. Clinical symptoms of TW-1 to TW-36 strain isolates.**

| sequence type | TW-1 | TW-2 | TW-3 | TW-4 | TW-5 | TW-6 | TW-7 | TW-8 | TW-9 | TW-10 | TW-11 | TW-12 | TW-13 | TW-14 | TW-15 | TW-16 | TW-17 | TW-18 | TW-19 | TW-20 | TW-21 | TW-22 | TW-23 | TW-24 | TW-25 | TW-26 | TW-27 | TW-28 | TW-29 | TW-30 | TW-31 | TW-32 | TW-33 | TW-34 | TW-35 | TW-36 | Sum |
|---|---|---|---|---|---|---|---|---|---|---|---|---|---|---|---|---|---|---|---|---|---|---|---|---|---|---|---|---|---|---|---|---|---|---|---|---|---|
| Fever | 172 | 3 | 1 | 9 | 9 | 6 | 6 | 6 | 45 | 16 | 1 | 2 | 3 | 1 | 1 | 11 | 2 | 5 | 49 | 5 | 4 | 87 | 3 | 11 | 3 | 5 | 5 | 3 | 3 | 1 | 2 | 2 | 1 | 1 | 1 | 3 | 488 |
| Eschar | 9 | 2 |  | 4 | 3 | 3 | 2 | 3 | 14 | 6 |  | 1 | 2 |  |  | 5 | 1 | 3 | 16 | 2 | 3 | 42 | 3 | 4 | 2 | 2 | 2 | 3 | 2 | 1 | 2 |  | 1 | 1 | 1 | 1 | 136 |
| Rashes | 32 | 2 | 1 | 2 | 1 | 3 | 2 |  | 30 | 6 |  | 1 | 2 |  | 1 | 6 |  |  | 18 | 2 | 1 | 11 |  | 4 | 2 |  |  |  | 2 | 1 | 1 | 2 | 1 | 1 |  | 2 | 138 |
| Headache | 45 |  | 3 | 3 | 2 | 2 | 2 | 3 | 12 | 6 | 1 |  | 2 |  |  | 3 |  | 3 | 21 | 2 | 1 | 33 | 1 | 3 | 3 | 1 | 1 | 2 | 2 | 1 |  |  |  |  |  | 2 | 156 |
| Lymphadenopathy | 17 | 2 |  |  |  | 1 | 1 | 1 | 5 | 1 |  | 1 | 1 |  |  | 3 |  |  | 6 |  | 1 | 6 |  | 2 | 1 |  |  |  | 3 |  | 1 |  |  |  |  |  | 52 |
| Abdominal pain | 5 |  |  | 1 |  |  |  |  | 3 |  |  |  |  |  |  | 1 |  |  |  | 1 |  |  |  | 1 |  |  |  |  |  |  |  |  |  |  |  |  | 14 |
| Diarrhea | 6 |  |  | 1 |  |  |  | 1 |  |  |  |  |  |  |  |  |  |  | 1 |  |  | 1 |  | 1 |  |  |  |  |  |  |  |  |  |  |  |  | 11 |
| Vomiting | 3 |  |  |  |  |  |  |  |  |  |  |  |  |  |  |  |  |  | 1 |  |  |  |  |  |  |  |  |  |  |  |  |  |  |  |  |  | 5 |
| Malaise | 8 |  |  |  |  |  |  | 2 | 1 | 1 |  |  |  |  |  |  |  |  | 5 |  |  | 4 |  | 3 |  |  |  |  |  | 1 |  |  |  |  |  |  | 25 |
| Chills | 10 |  |  | 1 |  |  |  |  | 1 | 1 | 1 |  |  |  |  |  |  |  | 3 |  |  | 7 |  | 1 |  |  |  |  |  |  |  |  | 1 |  |  | 1 | 27 |
| Dyspnea | 3 |  |  |  |  |  |  |  |  | 1 |  |  |  |  |  |  |  |  | 3 |  |  |  |  |  |  | 1 |  |  |  | 1 |  |  |  |  |  |  | 8 |
| Cough | 8 |  |  |  |  |  |  |  |  | 1 |  |  |  |  |  |  |  |  | 2 | 1 |  | 4 |  | 1 |  | 1 |  |  |  |  |  |  |  |  |  |  | 18 |
| Sore throat | 2 |  |  |  |  |  |  |  | 1 | 1 | 1 |  |  |  |  |  |  |  |  |  |  | 6 |  |  |  | 1 |  |  | 1 |  |  |  |  |  |  |  | 12 |
| Myalgia | 10 |  |  |  |  | 1 |  | 1 | 2 | 1 |  |  |  |  |  |  |  |  | 7 |  |  | 7 | 1 | 2 | 1 | 1 |  |  | 1 |  |  | 1 |  |  | 1 |  | 35 |
| Arthralgia | 1 |  |  |  |  | 1 | 1 |  |  | 1 | 1 |  |  |  |  |  |  |  | 1 |  |  | 5 |  |  |  |  |  |  |  |  |  | 1 |  |  |  |  | 12 |
| Drowsiness | 2 |  |  |  |  |  |  |  |  |  |  |  |  |  |  |  |  |  | 2 |  |  |  |  |  |  |  |  |  |  |  |  |  |  |  |  |  | 2 |
| Sepsis | 2 |  |  |  |  |  |  |  |  |  |  |  |  |  |  |  |  |  |  |  |  | 1 |  | 1 |  |  |  |  |  |  |  |  |  |  |  |  | 4 |
| Pneumonia | 2 |  |  |  |  |  |  |  |  |  |  |  |  |  |  |  |  |  |  |  |  |  |  |  |  |  |  |  |  |  |  |  |  |  |  |  | 2 |
| Consciousness change | 1 |  |  |  |  |  |  |  |  |  |  |  |  |  |  |  |  |  |  |  |  | 1 |  |  |  |  |  |  |  |  |  |  |  |  |  |  | 2 |
| Liver dysfunction | 13 |  |  | 1 |  | 1 | 1 | 2 | 1 | 3 |  |  |  |  |  | 1 |  | 1 | 4 |  |  | 10 |  | 3 | 1 |  |  |  |  |  | 1 |  |  |  | 1 |  | 43 |
| Jaundice | 3 |  |  |  |  |  |  |  | 1 | 1 |  |  |  |  |  |  |  |  |  |  |  | 2 |  | 2 |  | 1 |  |  |  |  | 1 | 1 |  |  |  |  | 12 |
| Kidney dysfunction | 3 |  |  |  |  |  |  |  |  |  |  |  |  |  |  |  |  |  |  |  |  |  |  |  |  |  |  |  |  | 1 |  |  |  |  |  |  | 4 |
| Poor appetite | 1 |  |  |  |  |  | 1 |  |  |  |  | 1 |  |  |  |  |  |  | 1 |  |  | 3 |  |  |  |  |  |  |  |  |  |  |  |  |  |  | 6 |
| Conjunctivitis | 1 |  |  | 1 |  |  |  |  | 3 |  |  |  |  |  |  |  |  |  | 1 |  |  | 3 |  |  |  |  |  |  |  |  |  |  | 1 | 1 |  |  | 6 |
| sequence sum | 198 | 4 | 1 | 10 | 9 | 6 | 6 | 7 | 48 | 18 | 1 | 2 | 3 | 1 | 1 | 11 | 2 | 5 | 54 | 5 | 6 | 97 | 4 | 12 | 3 | 6 | 5 | 3 | 4 | 3 | 2 | 2 | 1 | 1 | 1 | 3 | 545 |

14, and TW-15 sequence types were isolated in the cold season (November to February), and their *tsa56* sequences are closely related to the northern area of the endemic region.

Fever, headache, and eschars are the most common symptoms of scrub typhus, essential in making the clinical diagnosis. Eschars are the most useful diagnostic clue, and patients without eschars might be misdiagnosed as a common cold. The presentation of eschars varies from 7% to 97% in different geographic regions [9], such as 87% in Japan [23], 7.4% in Bhutan [24], and some patients with no eschars develop severe multiorgan dysfunction syndrome [25]. The incidence of the syndrome is summarized in Table 4. We found 41.5% (208/501) of patients with eschars were clinically detected by physicians. Other severe syndromes, including pneumonitis, liver and kidney dysfunction, accounted for 9.0% (49/545). At present, clinical diagnosis mainly relies on patients' self-description and physicians' experience. Our study reveals that sequence type analysis of the TSA56 gene may provide valuable information for treatment.

*Leptotrombidium deliense* is a pivotal vector for summer scrub typhus in the southern area of endemicity. In addition, *L. scutellare* is a principal vector for transmission of winter-type scrub typhus found in Taiwanese offshore islands (Kinmen and Matsu Islands) [21] and in the northern area of endemicity that includes China [26], Japan [27], and South Korea [28]. Other harboring OT chiggers, including *L. akamushi*, *L. deliense*, *L. imphalum*, *L. kawamurai*, *L. pallidum*, *L. rubellum*, and *L. scutellare* have been detected in Taiwan. The positivity rate of TSA56-PCR reached 55.9% in these chiggers [21]. Additionally, the seropositivity rate was 43% among captured rodents [21]. Taiwan is located in the center of the tsutsugamushi triangle, harboring abundant rodents and migratory birds from the East Asia/Australasia Flyway that might promote host diversity, the expansion of dominant vectors, and spreading diverse genotypes of *O. tsutsugamushi* [29].

It has been found that mixed or coinfection may be incurred in patients by evidence of different pathogens existing in tissue specimens (eschar and whole blood) assayed by the sequencing of PCR clones [30,31]. In fact, we are interested in this important issue. However, no additional coinfection information, such as SFTSV coinfection or mixed genotypes of scrub typhus, was found in our data. Further in-depth research on the topic is required to extend our knowledge of coinfection.

We noted that TW-9 showed a ratio of 48 of 545 and 99.7% nucleotide sequence similarity to the Boryong strain (AM494475). In addition, only 1 of 545 isolates showed sequence similarity to the Kawasaki strain (M63383) listed in the TW-15 line (Table 1). Additionally, the TW-15 strain was similar to the Taguchi strain (AF173038) [31], with nucleotide sequence similarity reaching 99.37%, indicating that only 1 isolate was closely related to the Taguchi genotype among the 545 isolates. These results suggest that the Boryong and closely related TW-9 strains are prevalent in South Korea [32] and Taiwan, respectively. In addition, our results showed that the Taguchi genotype has not yet become the main prevalent strain in Taiwan. Taiwan is geographically close to South Korea. Surveillance and research efforts for scrub typhus are needed in the future.

The immunodominant 56 kDa surface protein (TSA56) is a major surface protein that contains hypervariable regions and exhibits remarkable sequence variation in different strains. Analysis of the genetic relationship of *Orientia* strains using DNA sequences of the TSA56 gene may have exaggerated the differences in the evolution of these strains. However, in the past few years, TSA56 gene of *O. tsutsugamushi* has been the target for phylogenetic analysis because of its sequence variation. It is known that the sequence diversity within housekeeping genes is very restricted, leading to the analysis of conserved housekeeping genes by multilocus sequencing (MLS), which might be required for surveillance of genetic phylogeny. Using housekeeping genes as an alternative approach to study the evolution and phylogeny of *O. tsutsugamushi* is required for future comparisons of the present results.

Serotyping has been used to classify a new isolate of *O. tsutsugamushi* based on reactivity with strain- or type-specific monoclonal antibodies or hyperimmune sera recognizing a specific motif on TSA56 from well-characterized strains [33]. However, serotyping often exhibits moderate cross-reactivity between the unidentified isolate and prototype strains. In contrast, genotyping of TSA56 is a promising approach to determine the molecular epidemiology of *O. tsutsugamushi*. Therefore, we sequenced the complete TSA56 gene of 545 isolates to analyze the genetic diversity of *O. tsutsugamushi* during the 2006–2016 surveillance in Taiwan. Nevertheless, for a better understanding of the correlation between serotypes and genotypes of *O. tsutsugamushi*, it would be valuable to extensively investigate the serotypes of *O. tsutsugamushi* among our isolates in the future.

At present, knowledge of the antigenic variation of the immunodominant protein TSA56 is crucial for the development of effective diagnostic tools and a vaccine [34–36]. The investigation of the geographical distribution of *O. tsutsugamushi* genotypes *provides* valuable *insights into* the *epidemiology and control* of scrub typhus. Currently, scrub typhus is no longer restricted to traditional endemic areas, and it can be caused by species other than *O. tsutsugamushi* [37–41]. Therefore, further investigation of the antigenic diversity and prevalence in local endemic areas needs to be continued not only for the epidemiological monitoring of scrub typhus but also for the improvement of diagnostic accuracy and vaccine development.

## Supporting information

**S1 Table. Strain identifiers and their accession numbers of *O. tsutsugamushi* strains in Taiwan during 2006–2016.**
(PDF)

## Author Contributions

**Conceptualization:** Hsiang-Fei Chen, Shih-Huan Peng, Kun-Hsien Tsai, Cheng-Fen Yang, Mei-Chun Chang, Yeou-Lin Hsueh, Chien-Ling Su, Ruo-Yu Wang, Pei-Yun Shu, Su-Lin Yang.

**Data curation:** Hsiang-Fei Chen, Shih-Huan Peng, Kun-Hsien Tsai, Cheng-Fen Yang, Mei-Chun Chang, Yeou-Lin Hsueh, Chien-Ling Su, Ruo-Yu Wang, Pei-Yun Shu, Su-Lin Yang.

**Formal analysis:** Hsiang-Fei Chen, Shih-Huan Peng, Kun-Hsien Tsai, Cheng-Fen Yang, Mei-Chun Chang, Yeou-Lin Hsueh, Chien-Ling Su, Ruo-Yu Wang, Pei-Yun Shu, Su-Lin Yang.

**Investigation:** Shih-Huan Peng, Cheng-Fen Yang, Mei-Chun Chang, Yeou-Lin Hsueh, Chien-Ling Su, Ruo-Yu Wang, Pei-Yun Shu, Su-Lin Yang.

**Methodology:** Hsiang-Fei Chen, Shih-Huan Peng, Mei-Chun Chang, Yeou-Lin Hsueh, Chien-Ling Su, Ruo-Yu Wang.

**Project administration:** Pei-Yun Shu.

**Software:** Shih-Huan Peng, Cheng-Fen Yang, Mei-Chun Chang.

**Supervision:** Kun-Hsien Tsai, Su-Lin Yang.

**Validation:** Hsiang-Fei Chen, Shih-Huan Peng, Kun-Hsien Tsai, Mei-Chun Chang, Yeou-Lin Hsueh, Pei-Yun Shu, Su-Lin Yang.

**Visualization:** Mei-Chun Chang, Yeou-Lin Hsueh, Ruo-Yu Wang, Pei-Yun Shu.

**Writing – original draft:** Hsiang-Fei Chen, Shih-Huan Peng, Cheng-Fen Yang, Mei-Chun Chang, Yeou-Lin Hsueh, Chien-Ling Su, Pei-Yun Shu, Su-Lin Yang.

**Writing – review & editing:** Kun-Hsien Tsai, Pei-Yun Shu, Su-Lin Yang.

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
