## [Decision Letter · Decision Letter 0]

19 Jan 2022

Dear Dr. Yang,

Thank you very much for submitting your manuscript "Molecular epidemiology of scrub typhus in Taiwan during 2006-2016" for consideration at PLOS Neglected Tropical Diseases. As with all papers reviewed by the journal, your manuscript was reviewed by members of the editorial board and by several independent reviewers. The reviewers appreciated the attention to an important topic. Based on the reviews, we are likely to accept this manuscript for publication, providing that you modify the manuscript according to the review recommendations. 

The authors should address the issues raised by reviewers and modified the text accordingly for publication.

Sincerely,

Nam-Hyuk Cho

Deputy Editor

Jeanne Salje

Deputy Editor

The authors should address the issues raised by reviewers and modified the text accordingly for publication.

Reviewer's Responses to Questions

**Key Review Criteria Required for Acceptance?**

**Methods**

-Are the objectives of the study clearly articulated with a clear testable hypothesis stated?

-Is the study design appropriate to address the stated objectives?

-Is the population clearly described and appropriate for the hypothesis being tested?

-Is the sample size sufficient to ensure adequate power to address the hypothesis being tested?

-Were correct statistical analysis used to support conclusions?

-Are there concerns about ethical or regulatory requirements being met?

Reviewer #1: The methods are adequate to address the issue of genetic diversity of Orientia tsutsugamushi and isolates obtain from persons with scrub typhus in Taiwan.

The 56 kDa surface protein contains four hydrophilic hypervariable regions that are the apparent mechanism of antigenic diversity of Orientia tsutsugamushi. A better understanding of the evolution and phylogeny of Orentia tsutsugamushi would be better analyzed with conserved housekeeping genes.

Reviewer #2: The methods in this study are suitable.

**Results**

-Does the analysis presented match the analysis plan?

-Are the results clearly and completely presented?

-Are the figures (Tables, Images) of sufficient quality for clarity?

Reviewer #1: The establishment of 545 isolates Orientia tsutsugamushi from confirmed cases of scrub typhus and sequencing of the 56 kilodalton surface protein gene in all of them is a major achievement.

The criterion of 98% or greater sequence similarity as determining a distinct sequence type seems arbitrary. What is the rationale for this percent.

Lines 214-215: Liver dysfunction is listed as a symptom. Measurement of liver function is determined by laboratory tests and is not a patient complaint. It is unlikely that these patients had hepatic dysfunction; it is much more likely that they manifested hepatic injury by elevated transaminase enzymes. Was a measurement of hepatic function such as serum ammonium concentration or intrahepatic cholestasis identified?

Reviewer #2: The results in this study are good and suitable.

**Conclusions**

-Are the conclusions supported by the data presented?

-Are the limitations of analysis clearly described?

-Do the authors discuss how these data can be helpful to advance our understanding of the topic under study?

-Is public health relevance addressed?

Reviewer #1: 1. On lines 290 and 296 the issue of antigenic variation is raised. The data in the paper are gene sequences, i.e. genotypes. It is very likely that the most important antigenic variation of Orientia tsutsugamushi is determined by the four hypervariable regions of the 56 kDa surface protein. Knowledge of the genotype does not provide any information about the serotypes, which are very likely highly variable within the Karp group genotype. The reviewer is unaware of any correlation between serotypes and genotypes of Orientia tsutsugamushi. The situation is not clearly stated in this manuscript.

2. Lines 249-250: There is no evidence that animal hosts of the chiggers determine variation in Orientia tsutsugamushi. Rodents are hosts of the chiggers but are not hosts of Orientia tsutsugamushi. Although rodents are infected with Orientia tsutsugamushi and chiggers can become infected while feeding on an infected rodent, these chiggers do not transmit the bacteria transovarially. Chiggers are the only true host of Orientia tsutsugamushi. Animals are dead-end hosts.

3. Lines 272-275: Are the severe manifestations in patients infected with TW-1 and TW-2 statistically significantly different from the manifestations in other strains? Are the mild syndromes observed in patients with TW-3, TW-4, TW-5, and TW-17 statistically different compared with patients infected with other strains?

Reviewer #2: The conclusions are reasonable and suitable.

**Editorial and Data Presentation Modifications?**

Reviewer #1: I believe that the authors should address the issues that I have raised under conclusions, results, and methods.

Reviewer #2: Major Revision.

Please see "Summary and General Comments".

**Summary and General Comments**

Reviewer #1: This manuscript presents outstanding data and merits publication.

Reviewer #2: Authors studied the tsa56 genotypes of 545 O. tsutsugamushi clinical isolates throughout Taiwan, including offshore islands, from 2006 to 2016 to elucidate the molecular epidemiology of scrub typhus in Taiwan and data indicate the widespread presence of tsa56 genotypes closely related to Thailand and Korean strains and the presence of the unique endemic strains TW-12, TW-22, TW-29, and TW-36 in Taiwan. 

1. Several reports reported mixed (or co) infection of different genotypes in a patient in SE and East Asia 

(1. Emerg Infect Dis. 2018 Aug;24(8):1520-1523. doi: 10.3201/eid2408.171622. Dual Genotype Orientia tsutsugamushi Infection in Patient with Rash and Eschar, Vietnam, 2016. 

2. Am. J. Trop. Med. Hyg., 99(2), 2018, pp. 287–290. doi:10.4269/ajtmh.18-0088

Mixed Infection with Severe Fever with Thrombocytopenia Syndrome Virus and Two Genotypes of Scrub Typhus in a Patient, South Korea, 2017).

Do you also find mixed (or co) infection different in single patient in your study? If you find this, please also put this data in your study.

2. Jeju island, South Korea is close to Taiwan and Boryong and Taguchi genotypes of O. tsutsugamushi were found in a patient, Jeju Island, South Korea 

(Am. J. Trop. Med. Hyg., 99(2), 2018, pp. 287–290. doi:10.4269/ajtmh.18-0088 Mixed Infection with Severe Fever with Thrombocytopenia Syndrome Virus and Two Genotypes of Scrub Typhus in a Patient, South Korea, 2017). 

Authors showed that most isolates belonged to the Karp (49.9%; 272/545) genotype. 

Did you also find Boryong and Taguchi genotypes in your study? 

If you do not find these types, please give some opinion on the difference between Taiwan and Jeju Island, South Korea.

PLOS authors have the option to publish the peer review history of their article (what does this mean?). If published, this will include your full peer review and any attached files.

Reviewer #1: No

Reviewer #2: Yes: LEE, KEUN HWA

Figure Files:

Data Requirements:

Reproducibility:

References

---

## [Decision Letter · Decision Letter 1]

15 Mar 2022

Dear Dr. Yang,

Thank you very much for submitting your manuscript "Molecular epidemiology of scrub typhus in Taiwan during 2006-2016" for consideration at PLOS Neglected Tropical Diseases. As with all papers reviewed by the journal, your manuscript was reviewed by members of the editorial board and by several independent reviewers. The reviewers appreciated the attention to an important topic. Based on the reviews, we are likely to accept this manuscript for publication, providing that you modify the manuscript according to the review recommendations. 

One of the reviewers insisted a little more discussion on the genotype diversity of O. tsutsugamushi and potential co-infection with SFTSV in Taiwan.

Please respond and discuss these issues.

Sincerely,

Nam-Hyuk Cho

Deputy Editor

Jeanne Salje

Deputy Editor

One of the reviewers insisted a little more discussion on the genotype diversity of O. tsutsugamushi and potential co-infection with SFTSV in Taiwan.

Please respond and discuss these issues.

Reviewer's Responses to Questions

**Key Review Criteria Required for Acceptance?**

**Methods**

-Are the objectives of the study clearly articulated with a clear testable hypothesis stated?

-Is the study design appropriate to address the stated objectives?

-Is the population clearly described and appropriate for the hypothesis being tested?

-Is the sample size sufficient to ensure adequate power to address the hypothesis being tested?

-Were correct statistical analysis used to support conclusions?

-Are there concerns about ethical or regulatory requirements being met?

Reviewer #1: Methods are acceptable

Reviewer #2: YES

**Results**

-Does the analysis presented match the analysis plan?

-Are the results clearly and completely presented?

-Are the figures (Tables, Images) of sufficient quality for clarity?

Reviewer #1: The results are acceptable

Reviewer #2: YES

**Conclusions**

-Are the conclusions supported by the data presented?

-Are the limitations of analysis clearly described?

-Do the authors discuss how these data can be helpful to advance our understanding of the topic under study?

-Is public health relevance addressed?

Reviewer #1: The conclusions are acceptable

Reviewer #2: YES

**Editorial and Data Presentation Modifications?**

Reviewer #1: Nine

Reviewer #2: .

**Summary and General Comments**

Reviewer #1: The current form of the manuscript is excellent

Reviewer #2: Below is my comment. 

But, I cannot find my comment in the revision manuscript. 

1. Could you highlight in your comment or reply to it. 

--- 

Authors studied the tsa56 genotypes of 545 O. tsutsugamushi clinical isolates throughout Taiwan, including offshore islands, from 2006 to 2016 to elucidate the molecular epidemiology of scrub typhus inTaiwan and data indicate the widespread presence of tsa56 genotypes closely related to Thailand and Korean strains and the presence of the unique endemic strains TW-12, TW-22, TW-29, and TW-36 in Taiwan. 

1. Several reports reported mixed (or co) infection of different genotypes in a patient in SE and East Asia 

(1. Emerg Infect Dis. 2018 Aug;24(8):1520-1523. doi: 10.3201/eid2408.171622. Dual Genotype Orientia tsutsugamushi Infection in Patient with Rash and Eschar, Vietnam, 2016. 

2. Am. J. Trop. Med. Hyg., 99(2), 2018, pp. 287–290. doi:10.4269/ajtmh.18-0088

Mixed Infection with Severe Fever with Thrombocytopenia Syndrome Virus and Two Genotypes of Scrub Typhus in a Patient, South Korea, 2017).

Do you also find mixed (or co) infection different in single patient in your study? If you find this, please also put this data in your study.

2. Jeju island, South Korea is close to Taiwan and Boryong and Taguchi genotypes of O. tsutsugamushi were found in a patient, Jeju Island, South Korea 

(Am. J. Trop. Med. Hyg., 99(2), 2018, pp. 287–290. doi:10.4269/ajtmh.18-0088 Mixed Infection with Severe Fever with Thrombocytopenia Syndrome Virus and Two Genotypes of Scrub Typhus in a Patient, South Korea, 2017). 

Authors showed that most isolates belonged to the Karp (49.9%; 272/545) genotype. 

Did you also find Boryong and Taguchi genotypes in your study? 

If you do not find these types, please give some opinion on the difference between Taiwan and Jeju Island, South Korea.

---

2. Cases were detected in the Middle East and South America and also reported on Chiloé Island in southern Chile (N Engl J Med 2016; 375:954-961)

Could you put this (also reference(s)) in line 35 and 36? 

3. In line 91, please write the full name of “qPCR”

4. In line 129, please put more information about “Zeiss” such as the name of the city and country.

5. In line 140, please put more information about “QIAGEN” such as the name of the city and country.

6. In line 150, please put more information about “MEGA version 7” such as the name of the city and country.

7. In line 307 and 310, please write italics about “O. tsutsugamushi”.

PLOS authors have the option to publish the peer review history of their article (what does this mean?). If published, this will include your full peer review and any attached files.

Reviewer #1: Yes: David H Walker MD

Reviewer #2: No

Figure Files:

Data Requirements:

Reproducibility:

References

---

## [Editor Report · Decision Letter 2]

29 Mar 2022

Dear Dr. Yang,

We are pleased to inform you that your manuscript 'Molecular epidemiology of scrub typhus in Taiwan during 2006-2016' has been provisionally accepted for publication in PLOS Neglected Tropical Diseases.

Best regards,

Nam-Hyuk Cho

Deputy Editor

Jeanne Salje

Deputy Editor

---

## [Editor Report · Acceptance letter]

25 Apr 2022

Dear Dr. Yang,

We are delighted to inform you that your manuscript, "Molecular epidemiology of scrub typhus in Taiwan during 2006-2016," has been formally accepted for publication in PLOS Neglected Tropical Diseases.

Best regards,

Shaden Kamhawi

co-Editor-in-Chief

Paul Brindley

co-Editor-in-Chief
